

**Measurement Report: Comparative Analysis of Fluorescing African Dust Particles in Spain**
**and Puerto Rico**
Bighnaraj Sarangi[1], Darrel Baumgardner[2], Ana Isabel Calvo[3], Benjamin Bolaños-Rosero[4], Roberto
Fraile[3], Alberto Rodríguez-Fernández[5], Delia Fernández-González[5], Carlos Blanco-Alegre[3], Cátia
Gonçalves[3], Estela D. Vicente[3], Olga L. Mayol Bracero[1,6]
*[1]Department of Environmental Sciences, University of Puerto Rico – Río Piedras Campus, San Juan, Puerto Rico, USA*
*[2]Droplet Measurement Technologies, LLC, Longmont, Colorado, USA*
*[3]Department of Physics, Universidad de León, 24071 León, Spain*
*[4]Department of Microbiology and Medical Zoology, School of Medicine, University of Puerto Rico –Medical Sciences*
*Campus, San Juan, Puerto Rico, USA*
*[5]Department of Biodiversity and Environmental Management, University of León, Spain*
*[6]Now at Environment and Climate Sciences Department, Brookhaven National Laboratory, Upton, New York, USA*
Correspondence: Bighnaraj Sarangi (bighnarajsarangi1986@gmail.com)



**Abstract**

Measurements during episodes of African dust, made with two Wideband Integrated Bioaerosol Spectrometers (WIBS), one on the northeastern coast of Puerto Rico and the other in the city of León, Spain, show unmistakable, bioaerosol-like fluorescing aerosol particles (FAP) that can be associated with these dust episodes. The Puerto Rico events occurred during a major incursion of African dust during June 2020. The León events occurred in the late winter and spring of 2022 when widespread, elevated layers of dust inundated the Iberian Peninsula. Satellite and back trajectory analyses confirm that dust from Northern Africa was the source of the particles during both events. The WIBS measures the size of individual particles in the range from 0.5 µm to 30 µm, derives a shape factor and classifies seven types of fluorescence from the FAP. In general it is not possible to directly determine the specific biological identity from fluorescence signatures, however, measurements of these types of bioaerosols in laboratory studies allow us to compare ambient fluorescence patterns with whole microbial cells measured under controlled conditions. Here we introduce some new metrics that offer a more quantitative approach for comparing FAP characteristics derived from particles measured under different environmental conditions. The analysis highlights the similarities and differences at the two locations and reveals differences that can be attributed to the age and history of the dust plumes, e.g., the amount of time that the air masses were in the mixed layer and the frequency of precipitation along the air mass trajectory.

Keywords: African Dust, Bioaerosols, Fluorescing Aerosol Particles, WIBS, fluorescing particle finger prints.



## 1.0    Introduction

Fungal spores, spread by air currents, are some of the most abundant components of the bioaerosol population (Després et al., 2012). Their presence in the atmosphere has been linked to the formation of cloud condensation and ice nuclei (CCN and IN), thus playing a key role in the hydrological cycle (Huffman et al., 2013; Woo et al., 2018; Lawler et al., 2020). However, they also may have a negative environmental impact as many of them are described as important phytopathogens and constitute a hazard to animal and human health (Fröhlich-Nowoisky et al., 2016). Although there is a large fungal biodiversity the most abundant taxa are Ascomycetes and Basidiomycetes (Fröhlich-Nowoisky et al., 2009; Dietzel et al., 2019). The airborne spore load varies depending on the location and the season since such loading is closely linked to the vegetation and the meteorological conditions (Kasprzyk et al., 2015; Grinn-Gofroń et al., 2019; Anees-Hill et al., 2022; Rodríguez-Fernández et al, 2023). Nevertheless, the annual airborne dynamics can be altered by extreme weather phenomena such as thunderstorms, frontal systems or dust intrusions (Wu et al., 2004; Pulimood et al., 2007).

Dust intrusions are especially important because these long-distance, spore transport events allow them to colonize new environments (Rodríguez-Arias et al., 2023). In fact, some of these African dust (AD) events have been related to important environmental hazards such as decline of Caribbean coral reefs (Shinn et al., 2000; Hallegraeff et al., 2014). Agglomeration processes may also occur during these dust events, facilitating the adhesion of particulates with small diameters (40-90 nm) onto the surfaces of the pollen grains and spores.  Gravitational coagulation has been identified as the most likely mechanism of deposition on particle surfaces of about 20 μm (pollen grains) during long distance transport (Choël et al., 2022); however, other mechanisms of particle scavenging should not be underestimated. Fungal spores typically carry an electrostatic charge due to the complex chemical composition of their cell walls (Hannan, 1961; Leach, 1976; Feofilova, 2010; Wargenau et al., 2011). The differences in electrostatic charges can cause other particles to bind to the spore surface (Visez et al., 2020). The agglomeration process may increase the allergenic potential of airborne spores and pollen due to chemical reactions, increasing the health risk for allergy sufferers (Sénéchal et al., 2015).

The arid regions of Northern Africa are some of the largest sources of desert dust in the world. These regions emit about 800 Tg yr$^{-1}$, corresponding to approximately 70% of the annual, global dust loading (Prospero et al., 2014; Ryder et al., 2019). A large fraction (~182 million tons yr$^{-1}$) of these emissions moves westward, ~2500 km across the Atlantic Ocean, extending in a continuous AD plume over the Caribbean basin (Yu et al., 2015). Similarly, the transport of AD over Europe has a clear seasonality whereby such events are more frequent from late autumn to early winter (February to June) (Escudero et al., 2005). With climate change contributing to further desertification, not only on the Africa continent, but in Asia and other parts of the world, dust incursion events will likely increase in intensity and duration in the coming decades.

The majority of studies that have evaluated the transport of bioaerosols by AD have used samplers that captured the particles on substrates that were subsequently analyzed in the laboratory under a microscope or using Deoxyribonucleic acid (DNA) analysis. These analysis methodologies are the most robust for identifying specific taxa of biological spores; however, online techniques offer the advantage of larger sample sets that can be evaluated in much higher temporal resolution. Techniques that use Ultra-violet Laser Induced Fluorescence (UV-LIF), such as the Wideband Integrated Bioaerosol Spectrometer (WIBS), provide detailed information on the size, shape and fluorescence intensity of individual particles in real time (Kaye et al., 2004). The recent investigation by Morrison et al. (2020) employed a WIBS that was situated at the Sao Vicente Cape Verde Atmospheric Observatory, off the west coast of central Africa, measuring continuously from September 2015 to August 2016.  Their measurements found strong seasonal changes in absolute concentrations of fluorescing aerosol particles (FAP) with significant enhancements during winter due to the strong



island inflow of air masses originating from the African continent. Their results indicate that the relative contribution of bioaerosol material in dust transported across the tropical Atlantic throughout the year is relatively uniform, consisting mainly of mixtures of dust and bacteria and/or bacterial fragments. They support their conclusions by comparing the WIBS measurements with those from a Laser Ablation Aerosol Particle Time of Flight mass spectrometer (LAAP-ToF). The latter measurements show a high correlation between particles with mixed bio-silicate mass spectral signatures and UV-LIF bio-fluorescent signatures, leading to the conclusion that the FAP concentrations are dominated by these mixtures.

The measurements reported here, in our current study, complement those of Morrison et al. (2020) with results from locations farther down-wind than their study site of Cape Verde. Our objectives are to *1) expand the database of real time measurements related to long-range transported African dust and the FAP associated with these events, 2) evaluate the relative changes in the multi-faceted patterns of fluorescing particles, measured with the UV-LIF technique, as they relate to the air mass sources and ages, 3) introduce new metrics, unique to the UV-LIF technique that provide additional quantification of the FAP properties and 4) compare the real time fluorescence signatures to those bioaerosols measured with off-line techniques.*

## 2.0    Measurement locations, sensor description and analysis methodology
## 2.1    Study zones

The Caribbean measurement site is the Cape San Juan (CSJ) atmospheric observatory (18°22.85'N, 65°37.07'W and 60 m, asl) located  on the most northeastern point on the coast of Puerto Rico (PR). The European measurements were made at the University of León, León, Spain, located in the northwest region of the Iberian Peninsula (42° 36′ N, 05° 35′W and 838 m, asl). Cape San Juan is a remote, coastal research site managed by the Atmospheric Chemistry and Aerosols Research (ACAR) group at the University of Puerto Rico – Rio Piedras Campus (UPR-RP). This measurement site has been frequently used for sampling aerosols of non-anthropogenic origin (Novakov et al., 1997; Mayol-Bracero et al., 2001; Allan et al., 2008) because the predominate airflow is from the northeast and the particles are typically those generated from the ocean. i.e. sea salt, non-sea salt sulfates and organic carbon (Allan et al., 2008)    Furthermore, CSJ is also a recognized site for the World Meteorological Organization's Global Atmospheric watch (WMO GAW) (Andrew et al., 2019) and the National Aeronautics and Space Administration (NASA) network for AERONET, PANDORA, and MPLNET.

The city of León is located in the northwest of the Iberian Peninsula. The climate has Mediterranean maritime as well as continental features (Calvo et al., 2018). The sampling site is on the roof of the Faculty of Veterinary, 15 m above ground level, at the León University Campus. The university is located in the northeast suburban region of the city, which is largely devoid of local industrial emissions, although there are daily anthropogenic emissions from vehicular traffic whose organic compounds, like polyaromatic hydrocarbons (PAH), will fluoresce and need to be removed from the evaluation as non-bioaerosols, as discussed below.

## 2.2    Data sets and sources

The data used in the present study comes from several sources of in-situ and remote sensor measurements, as well as air mass back trajectories derived from archived meteorological data. Table I lists the data sets that have been evaluated, their sources and the parameters that were extracted. The primary source of particle information comes from the Wideband Integrated Bioaerosol Spectrometer (WIBS) since the main focus of our study is on the FAP that is being transported by AD. Ancillary information about the origins of the air masses, complementary measurements of the particle optical





properties and the state of the local environments, e.g., meteorology, are included in order to better
understand the impact of the AD intrusions.

Given the importance of the WIBS measurements, the following section focuses on the WIBS's
measurement principles, limitations and uncertainties, the filtering necessary to minimize artifacts in
the data, the corrections applied for dead-time losses and the parameters that are derived that provide
tracking of the unique patterns that are found in the particle properties.

Table I

Data sets used in the Analysis


| Data Set Description | Data Source | Extracted Parameters | Measurement Sites |
|---|---|---|---|
| Single particle aerosol properties | WIBS-V[1] | Aerosol particle equivalent optical diameter, 0.5 – 30 μm, autofluorescence, Asphericity factor, non-FAP and FAP number concentrations. | PR and LUC |
| Fog properties | FM-120[1] | Fog droplet equivalent optical diameter, droplet number concentration, liquid water content. | LUC |
| Aerosol Particle mass | MET-1[2], | Mass concentration in particles with aerodynamic diameter < 10 μm (50% cut size) | LUC |
| Filter samples | Hirst spore trap (VPPS 2000, Burkard)[3] | Morphological identification of fungal spore and pollen taxa. | PR and León |
| Aerosol optical properties | Aethalometer[4], AERONET Sun photometer[5] | 370 nm and 880 nm extinction coefficient, Multi-wavelength optical depth | PR |
| Local environments state parameters and radiation | Meteorological weather stations[6] | Temperature, humidity, pressure, wind speed and direction and visibility | PR, LUC |
| MERRA-2 | Modern-Era Retrospective analysis for Research and Applications version 2[7] | Column mass density of aerosol components (black carbon, dust, sea salt, sulfate, and organic carbon), surface mass concentration of aerosol components, and total extinction (and scattering) aerosol optical thickness (AOT) at 550 nm | PR, LUC |
| Air mass back trajectories | Hysplit[8] | Location and meteorology at hourly intervals | PR, LUC |


---

[1] Manufactured by Droplet Measurement Technologies, LLC, Longmont, CO
[2] Manufactured by Met One instruments, Grants Pass, OR
[3] Manufactured by Lanzoni,Bologna, Italy and Burkard Scientific Ltd, Uxbridge, UK
[4] Manufactured by McGee Scientific Inc, Berkely, CA
[5] Manufactured by CIMEL Electronique, Paris, France
[6] Manufactured by Vaisala Instruments and Davis Instruments, Hayward, CA, USA
[7] https://disc.gsfc.nasa.gov/datasets/M2T1NXAER_5.12.4/summary
[8] https://www.ready.noaa.gov/index.php



### 2.3 Wideband Integrated Bioaerosol Spectrometer (WIBS)

#### 2.3.1 Principles of operation, uncertainties and limitations

The WIBS measurement principles are based on Ultraviolet light-induced fluorescence (UV-LIF) (Kaye et al., 2005; Stanley et al., 2011). The current model, the WIBS-V differs from earlier models only in how the data are formatted and how deadtime losses are taken into account. The supplementary material describes the WIBS in greater detail, along with the specific algorithms used to filter and correct the measurements prior to analysis and interpretation. All WIBS models bring individual particles into the instrument with an internal pump and direct them through a collimated laser beam using aerodynamic focusing. The light scattered from each particle is used as a signal to trigger two xenon flash lamps, which activate sequentially, illuminating the particle as it leaves the laser beam with light filtered at 280 nm and 370 nm, respectively. Two detectors, one with a bandpass filter at 310-400 nm and the other with a 420-650 nm filter, receive light emitted by autofluorescence if there is material in the particle that is excited to fluoresce at one or both excitation wavelengths. The equivalent optical diameter (EOD) is derived from the light scattered by the particle as it transits the laser beam and an "asphericity/asymmetry factor" (AF) is derived from a quadrant detector that is illuminated by the forward scattered light from this same particle.

A number of naming conventions have been introduced in the literature over the years for labeling the fluorescence combinations that are possible with the WIBS measurements; however, they all agree on using FL1 (Channel A) and FL2 (Channel B) to denote signals from the excitation at 280 nm, emissions at 310-400 nm and 420-650 nm, respectively, and FL3 (Channel C) to indicate signals from excitation at 370 nm and emissions at 420-650 nm. As is often the case, the fluorescence from a single particle may be a combination of any two or three of these excitation/emission pairs, leading to as many as seven possible fluorescent types. Following the convention first proposed by Perring et al. (2015), we will label the seven fluorescence types as A, B, C, AB, AC, BC and ABC throughout the remainder of this presentation.

The two major sources of uncertainty are fluorescence artifacts and missed fluorescence signals due to deadtime. Both of these uncertainties, and steps taken to minimize or to correct for them, are discussed in greater detail in the supplemental material, as well as in previous publications (Calvo et al., 2018; Sarangi et al., 2022). In short, there are two types of fluorescence artifacts: 1) light detected by the fluorescence detectors that wasn't fluorescence from ambient particles and 2) light detected by the fluorescence detectors that is produced by non-biological materials.

If the chamber where particles are illuminated by the flash lamps is not cleaned after regular use, material may accumulate that will fluoresce, albeit at a fairly low level. Nevertheless, this fluorescence represents a source of background noise that needs to be quantified and removed from the signal produced by legitimate FAP. A second source of fluorescence artifacts is the light from the Xenon lamps themselves, a small fraction of which can leak through the filters in front of the fluorescence detectors since these filters are not 100% efficient at removing light at wavelengths outside their wave band. Non-biological materials, such as polycyclic aromatic hydrocarbons (PAH) or black carbon, which can also fluoresce when illuminated, are considered here as artifacts with respect to differentiating them from fluorescing bioaerosols (Gabey et al., 2013; Perring et al., 2015; Pöhlker et al., 2012; Toprak and Schnaiter, 2013). These artifacts cannot be completely removed from the analysis but can be minimized by removing from the processing any particles whose fluorescence falls below a preset threshold. As described in the supplementary material, we follow the methodology of Perring et al., (2015) and Morrison et al (2020) by creating daily frequency histograms of the FL1, FL2 and FL3 type FAPs and use a threshold that is the mode of the frequency distribution plus nine standard deviations ($9\sigma$) as the minimum threshold that has to be exceeded before a fluorescing event is accepted as valid.





The aforementioned uncertainty due to electronic deadtime is associated with the eight milliseconds
that is required to recharge after each Xenon lamp flash. During this period, if the lamps receive a
trigger signal, they will not discharge so if the particle passing through the chamber is an FAP, it will
not be identified as such since it won't be excited by the lamps. The WIBS registers the particle's
size but a statistical correction is needed to account for the fraction of particles each second that might
have been FAP but passed through the Xenon chamber during a "dead time". The supplementary
material discusses how this correction is derived.
**2.3.2   FAP features extracted from laboratory bioaerosol studies**
The FAP that were measured for the current study are assumed to be bioaerosols since we have taken
care to minimize artifacts; however, we are unable to *a priori* use the FAP properties to label the
particle as a specific type of bioaerosol, i.e. bacteria, fungal spores or pollen (BSP), to name the three
bioaerosol types most commonly found in the ambient environment (henceforth, we will group these
three types of bioaerosols and refer to them as BSP). We will take, instead, the same approach as in
Calvo et al. (2018) and refer to a specific FAP, for example, as "bacteria-like" or "fungi-like" when
a specific set of FAP metrics in the environmental measurements match the same metrics derived
from laboratory measurements.
We have reprocessed the data set that was used in the Hernandez et al. (2016) laboratory studies: 15
types of bacteria, 29 types of common fungal spores and 13 varieties of pollen, those typically found
in the natural environment.  Figure 1a shows a composite of the fluorescence type and EOD of the 57
different varieties of BSP. Figure 1b shows an example of these same varieties superimposed on a
composite of measured FAP for a non-dust day in PR. This illustrates how the environmental data
clusters by FAP type and EOD in patterns very similar to those formed from the laboratory
measurements. The color scale in Fig. 1b denotes how frequently during the two-day period the FAP
types and EODs fell within the different FAP Type vs EOD regions. In this example, although the
environmental FAPs fall in regions where the lab data show bacteria, fungal spores and pollen, quite
a few of the FAP were in the regions of FAP types C and AB where very few of the lab results were
found.

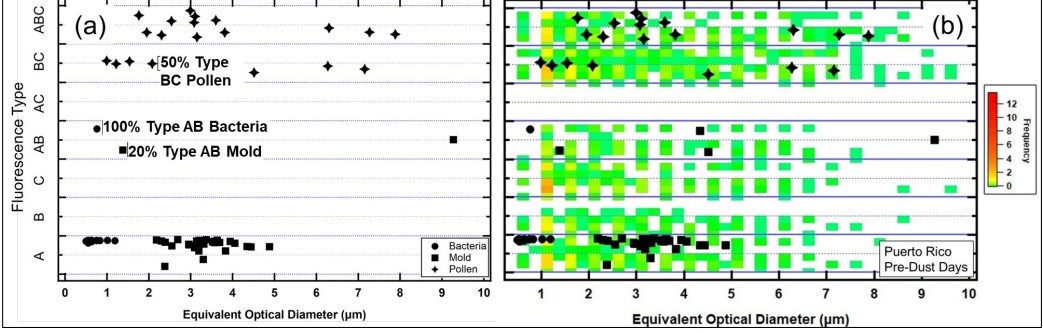

*Figure 1. a) A BSP map showing how 15 bacteria, 29 mold and 13 pollen, of different taxa were measured in the*
*laboratory by a WIBS, as a function of FAP type and EOD and b) the same BSP map with FAP measurements from*
*a non-dust day in Puerto Rico plotted using the same definitions for FAP type and EOD. The color scale denotes how*
*frequently during the two-day period the FAP types and EODs fell within the different regions. In this example,*
*although the environmental FAPs fall in regions where the lab data show bacteria, mold and pollen, quite a few of the*
*FAP were in the region of FAP types C and AB where very few of the lab results were found.*
Although this method of comparing lab BSP patterns with environmental FAP cannot be construed
as a quantitative way to relate the WIBS measurements directly to BSP taxa, the laboratory data





provides a reference data set to which we can compare the measured BSP maps and evaluate relative
changes in patterns related to the AD intrusions.
### 2.3.3  Working Hypothesis and Analysis Metrics
From our work, and from those of others, we have sufficient measurements to conclude that
fluorescence intensity, regardless of the BSP taxa, is too variable to be used as a FAP property that
can be unequivocally or unambiguously related to a bioaerosol type. Likewise, the asymmetry factor
can be used as a rough indicator of asphericity but cannot provide finer structural details. The
fluorescence emission intensity is a complex interaction between the uniformity of the excitation
radiation over the surface of the FAP, the orientation of the particle as it is exposed to the incident
light, the non-isotropic nature of the fluorescence emissions and fluorescence quenching by material
mixed with the FAP (Lakowisz, 2006). Adding to these uncertainties are the observations from
microscopic analysis that a significant fraction of bioaerosols in the natural environment are not
intact, i.e. they are fractured pieces that can still fluoresce but with less intensity and shapes
unrepresentative of a whole particle.
The specific pairs of excitation/emission wavelengths employed in the WIBS were originally selected
by Kaye et al., (2005) due to their responsiveness to tryptophan (280 nm/310-400 nm) and
nicotinamide adenine dinucleotide (NAD; 370 nm/420-650 nm). Given that these two fluorophores
are omnipresent in plant tissues and microbiological cells they are good fluorescent markers for
bioaerosols. Nevertheless, the aforementioned uncertainties prevent more definitive speciation of the
FAP without complementary analysis using samples captured on filters or substrates that can undergo
microscopic analysis and classification by human observers or more intensive DNA analysis.
The advantage and power of the WIBS is the high-resolution information that it extracts from
individual particles, information that provides a statistically large sample that describes the sizes,
shapes and fluorescence patterns of an ensemble of particles in air masses whose properties can
change over relatively short time periods. Hence, even though there are large variations in these
properties, particle by particle, over periods as short as five to ten minutes, tens of thousands of
particles in the size range of 0.5 to 30 µm can be analyzed. Not only are the average properties
important, but their variances also contain valuable information about the composition and potential
source of these particles.
The analysis methodology that is selected to evaluate the WIBS data needs to be tailored to the
specific questions that are being addressed. Much progress has been made in the past 10 years in the
use of cluster analysis to identify features of the FAP, which are indirectly related to the type of
bioaerosol (Robinson et al., 2013; Crawford et al., 2015; Morrison et al.,2020). In our study, however,
we are asking a different set of questions, where knowing the type of bioaerosol is not as important
as understanding how the FAPs are transported and their properties transformed while in AD plumes.
Hence, we take a more heuristic approach whereby we concentrate on evaluating the nine parameters
(size, shape and seven FAP types) that can be extracted from individual particles, and we use these
to address the following questions:
1. Are the FAP that are found within the AD plumes, which inundate the Caribbean and Iberian
Peninsula, internally or externally mixed with the dust particles?
2. What features of the FAP change from normal background conditions to periods when the
AD is present?
3. Can the observed changes in the FAP properties be physically linked to the air mass histories?
Starting with the assumption that the properties of aerosols in dust plumes will differ significantly
from those of aerosols in the local environments of PR and León, ***we hypothesize that 1) the***



*bioaerosols that are in the dust plumes will be a mixture of bioaerosol types that differ from those*
*found in the PR or León ambient environments and 2) the majority of FAP in the AD plumes will*
*be attached or mixed with dust particles.*
To provide answers to these questions, and to test our working hypotheses, we focus on how the size
distributions of number concentration, fluorescence intensity and shape factor differ within the
populations of non-FAPs and FAP. These differences, between the PR and León sites, before and
during AD events, can be quantified using comparisons of the size distribution metrics. These metrics
can be visualized by referring to Fig. 2 that shows example size distributions of the FAP Type ABC
fluorescence intensity (Fig. 2a), number concentration (Fig. 2b) and shape factor (Fig. 2c), before
(green curve), and during (brown) a dust intrusion. An EOD of 5 µm has been arbitrarily selected as
the threshold between "small" and "large" particles.

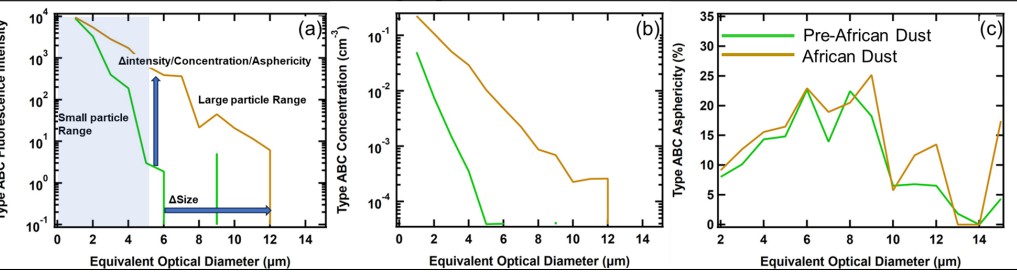

*Figure 2. Examples of size distributions of Type ABC fluorescing aerosol particles before (green curve) and during*
*(brown curve) a dust intrusion, highlighting the features that are used as metrics in the analysis methodology. (a)*
*Average fluorescence intensity as a function of EOD. The vertical and horizontal blue arrows highlight increases in*
*intensity and size, respectively, with the incursion of AD. The size distributions have been divided into "small particle"*
*(shaded) and "large particle". (b) average number concentration as a function of size and (c) Average asphericity as*
*a function of size*
The metrics that are derived from these size distributions and that we will use in our comparative
analysis are:
1. The change in fluorescence intensity, number concentration and shape factor of small
particles
2. The change in fluorescence intensity, number concentration and shape factor of large
particles
3. The change in the ratio between the concentration of small to concentration of large particles
4. The change in median diameter
From the examples shown in Fig. 2, for the FAP Type ABC, there is a significant increase in the
average fluorescence intensity and number concentration of small and large EOD particles with the
intrusion of AD; however, the shape factor size distributions are similar. As will be highlighted below,
these differences can be interpreted in the context of the relative mixture of FAP types and also how
these FAP are physically mixed with non-FAP.
**2.4    Complementary measurements**
Meteorological data, including temperature (°C), relative humidity (RH, %), rain (mm), pressure
(mb), wind speed (WS, m s$^{-1}$) and wind direction (°) were accessed from the weather station (Vaisala,
WXT 530) mounted on the top (30' from the ground) railing of an aluminum tower at the Puerto Rico
measurement site (CSJ).





The optical properties of the particles in PR were measured in situ with an aethalometer (Magee Scientific) and remotely with a sun photometer. The aethalometer derived the absorption coefficients from measurements of attenuations at 370 nm and 880 nm. The spectral Aerosol Optical Depth (AOD), Ångström exponent, and volume size distributions were accessed from the sun/sky CIMEL CE_318 Sun photometer that measures the direct solar irradiances with a field of view of approximately 1.2° and the sky radiances at spectral wavelengths of 340, 380, 440, 500, 675, 870, 1020, 1640 nm, respectively. The CIMEL Sun photometer at CSJ is a component of NASA Aerosol Robotic NETwork (AERONET) that provides long-term records of columnar aerosol optical characteristics (Holben et al., 1998) since 2004.

In addition to the particle mass (PM) measurements made with the PM Beta monitor at the Junta de Castilla and León air quality stations, an FM-120 fog monitor was operated in parallel with the WIBS in León. The FM-120, developed by Droplet Measurement Technologies LLC, measures the EOD of individual environmental particles from 2 - 50 μm. The FM-120 was originally developed to measure fog droplet properties; however, the measurements are not specific to fog and in the presence of dust particles will measure their size distributions but with a larger uncertainty because these particles will not be spherical.

Fungal spores and pollen were collected with Hirst samplers (Hirst, 1952) in PR and Leon where they were subsequently analyzed and classified by inspection under a microscope.

## 3.0 Results

Prior to delving into the details of the in situ WIBS measurements, we use remote sensing data to provide the complementary evidence for the large dust incursions on those days where the WIBS measured particle properties that were anomalous when compared to those normally encountered during the respective summer and spring seasons in PR and León.

### 3.1 Remote Sensing Observations

Satellite images from the Suomi National Polar-Orbiting Partnership (Suomi NPP, https://ncc.nesdis.noaa.gov/VIIRS/) show a high frequency of dust intrusions over the North Atlantic during the spring and summer of 2020. One of these events was an intense, widespread dust plume that was observed over the eastern North Atlantic, clearly originating from the Aftican Sahara region. The June 23 2020 satellite image, shown in Fig. S1, reveals a large region of dust over the Caribbean with another extensive layer of dust leaving northern Africa. This dust plume, which at some point had a size equivalent to the area of continental USA (around 8,000,000 km$^2$), impacted the Caribbean region and parts of South America, Central America, Gulf of Mexico, and the Southern USA from June 21 to July 1. On June 20, when the first dust pulse began to affect the Caribbean, a second dust layer was clearly seen leaving Africa (Fig. S1a), but smaller in extent than the first one (Yu et al., 2021). This second dust layer impacted the same area as the first plume from 26th of June to July 1. On June 22-23, PR received the leading edge of the dust plume followed by a second dust innundation on June 28-29 (Fig. S1a). This event has been reported by a number of research groups (Francis et al., 2020; Pu and Jin, 2021; Yu et al., 2021;Asutosh et al., 2022). According to Pu and Jin (2021), the meteorology behind this dust plume is unprecedented: the surface wind speed (the strongest since the previous 42 years) increased the dust emissions in Africa followed by an intensified African Easterly Jet (AEZ) moving the dust plume westward. Francis et al (2020) posit that the extreme dust event was caused by the development of a subtropical high-pressure system over northwest Africa that led to the strong north-easterlies that were sustained over the Sahara generating four days of continuous dust emissions. This dust event is also clearly seen from the Modern-Era Retrospective analysis for Research and Applications, Version 2 (MERRA-2), shown in Fig. 3a for June 23, 2020, which shows clearly the same patterns that were derived from the Suomi NPP satellite products (Fig. S1a).


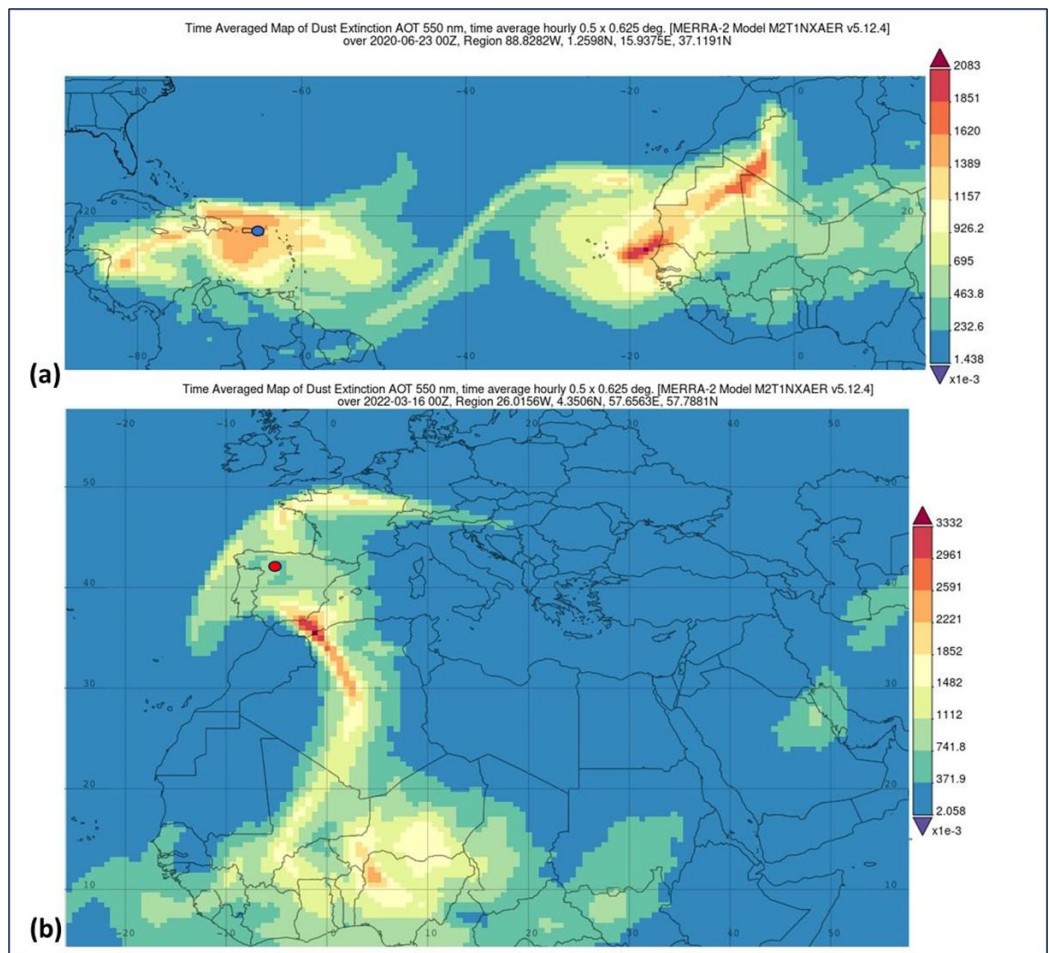

**Figure 3 The Aerosol Optical Thickness (AOT) at 500 nm, derived from the MERRA, show the air masses carrying**
**dust from the African continent over (a) Puerto Rico, on 23 June, 2020 and then (b) another plume traveling over the**
**Iberian Peninsula and Southern Europe on 16 March 2022. The blue and red markers indicate the locations of the**
**PR and Leon measurement sites, respectively.**

The Iberian Peninsula is also frequently inundated by Saharan dust outbreaks due to its proximity to
large dust-emitting areas of the Sahara and Sahel deserts and to the atmospheric dynamics and
meteorological conditions (Alastuey et al., 2016; Escudero et al., 2007; Querol et al., 2014; Rodríguez
et al., 2001). Previous studies reported that most of the outbreaks occur between spring and summer
when the dust transportation is regulated by the anticyclonic activities over the east or southeast of
the Iberian Peninsula (Lyamani et al., 2015; Rodríguez et al., 2001; Salvador et al., 2013). In winter,
Saharan dust intrusions are scarce and are usually dominated by the cyclonic activities over the west
or south of Portugal (Díaz et al., 2017; Rodríguez et al., 2001). However, in late winter, 2022, an
unprecedented dust storm impacted the Iberian Peninsula. The dust layer traveled over a large portion
of Europe, initially on 16-17 March 2022, followed by a secondary dust plume that covered an
extended region 27-30 March 2022. The satellite imagery obtained with the Suomi NPP clearly shows
the dust layer over the Iberian Peninsula on March 16 and 17 2022 (Figs S1b and c) and also seen in
the images derived from the MERRA-2 data (Fig. 3b)






## 3.2 WIBS Observations

The arrival of the AD over PR and León is reflected in large increases in the number concentration as seen in the time series of the FAP size distributions shown in Figs. 4a and b, respectively. In PR, the first dust intrusion is seen on June 21 (day of the year, DOY, 172) and then approximately six and a half days later the second AD layer arrives on June 27 (DOY 179). Likewise, in León the first AD incursion is detected by the WIBS on March 16 (DOY 74) and lasts for more than five days. This event was followed 10 days later by the second inundation on March 26 (DOY 84) lasting another five days.

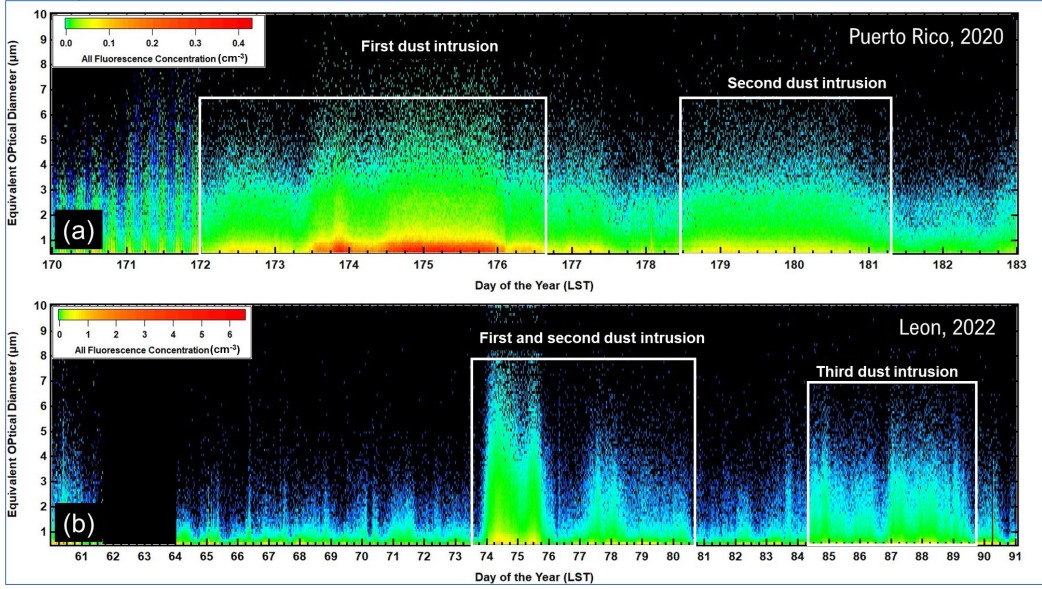

*Figure 4. a) Time series of the size distributions of FAP number concentrations measured at PR, Puerto Rico. The white boxes delineate the periods when the satellite and back trajectory analyses indicate that AD has arrived over the island, b) similar to (a) but for FAP concentrations measured in León, Spain.*

The influence of these dust incursions on the general aerosol population can be observed by the changes in particle asphericity, shown in the size distributions of the shape factor (percent asphericity) drawn in Fig. 5. These size distributions are of the non-FAP aerosols and show that the shape factor increases from quasi-spherical, i.e. shape factor < 10%, to > 30% during the periods of AD in PR and León.

The size distributions shown in Fig. 6 highlight the similarities and differences between the PR and León aerosol populations and illustrate how the arrival of the AD significantly changes how the non-FAP and FAP number concentrations vary with size. The PR and León distributions are drawn in black and green, respectively, solid lines for pre-dust events, dashed for dust intrusions. The pre-dust size distributions of non-FAP aerosol (Fig. 6a) are almost identical at both sites, with a small fraction of the León particle population larger than those in PR. The arrival of dust leads to almost two orders of magnitude increase in both the PR and León concentrations, over all sizes, and brings significant numbers of particles larger than 10 μm. For the non-FAP aerosols the relationship between concentration and size during the AD event is nearly the same for PR and León.

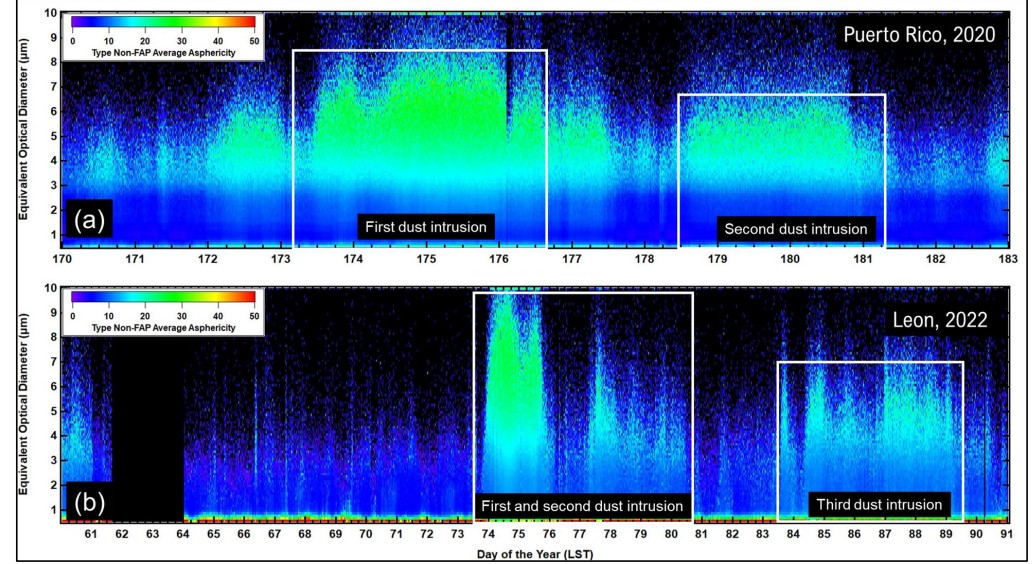

*Figure 5. a) Time series of the size distributions of non-FAP asphericity measured at PR, Puerto Rico. The white boxes*
*delineate the periods when the satellite and back trajectory analyses indicate that AD has arrived over the island, b)*
*similar to (a) but for non-FAP asphericity measured in León, Spain.*
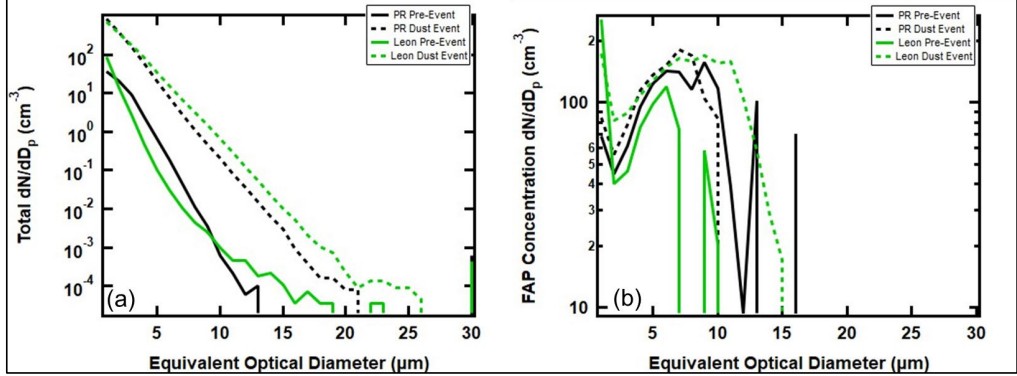

*Figure 6. Average size distributions in PR and León, before and during AD events for the (a) number concentrations*
*of the total aerosol population in the size range of the WIBS, b) number concentrations of only FAP.*
A comparison of the FAP size distributions (Fig. 6b) tells a very different story. Below 2 μm, the
León FAP concentrations exceed those in PR by about a factor of four; however, the PR FAP pre-
dust size distribution is much broader than the FAP in León, extending beyond 10 μm while the FAP
in León ends around 7 μm. The arrival of the dust does little to change the general shape of the PR
size distribution other than slightly narrowing it. In contrast, the León size distribution broadens
significantly out to 15 μm. This difference between PR and León offers the first clue that there is a
difference between PR and  León with respect to how FAP are mixed with non-FAP in the AD plumes
that inundate these two sites.
The average number concentration of non-FAP and FAP, the ratio of FAP to non-FAP concentrations,
and the median, equivalent optical diameters (MEOD) of non-FAP and FAP are bulk parameters that
are extracted from the size distributions and are shown in Fig. 7 for periods with no influence from
AD and those in the presence of dust. Whereas Figs. 6a and b only showed one period with no-dust





and one period with AD for PR and León, Fig. 7 includes the second periods of dust, for the two
locations, accompanied by periods before and after the dust intrusion. This more comprehensive data
set demonstrates that for both PR and León there are clear differences in the bulk parameters under
no-dust and dust conditions. The total and FAP number concentrations increase by an order of
magnitude in PR and León when the AD arrives, as compared to the no-dust periods (Figs. 7a and b).
The ratios of FAP to all particles (Fig. 7c) increase by a factor of two in PR and León under AD
conditions; however, the León FAP ratios are three times larger than PR in the presence of AD.
Likewise, although the arrival of dust in PR and León leads to increases in the average MEOD of all
particles and FAP (Fig. 7c and d), the increase in León is much more than in PR, 200-300% vs 30%,
respectively.

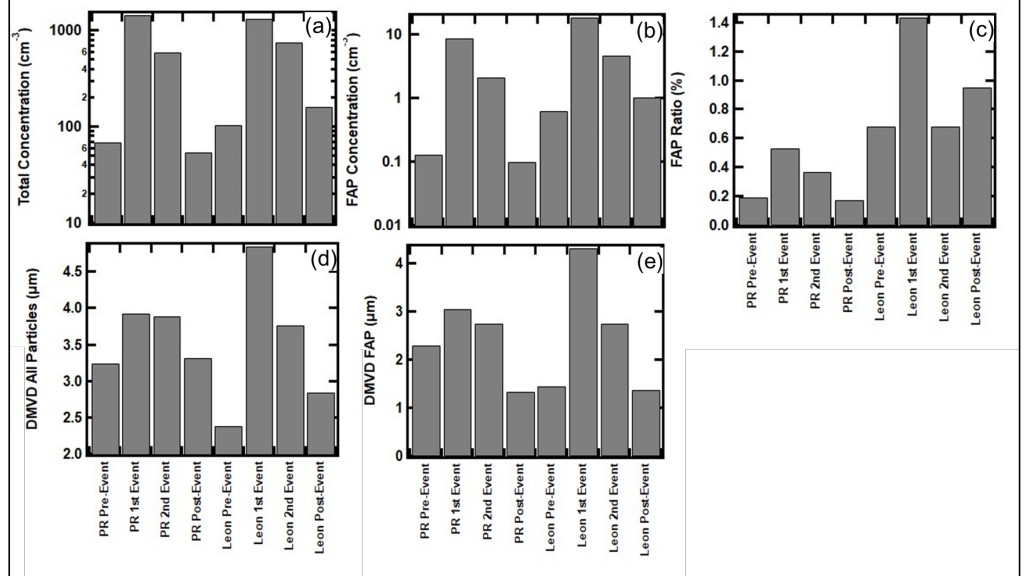

*Figure 7. Average values of derived parameters from WIBS measures before, during and after AD events. (a) Total*
*number concentrations, (b) Number concentrations for all FAP, (c) Ratio of all FAP to all particles, (d) Median volume*
*diameter (DMVD) of all particles between 0.5 and 30 μm and(e) DMVD of all FAP between 0.5 and 30 μm)*
Figure 8 takes a closer look at the FAP, stratifying them by the fluorescing types. The pre-dust event
aerosols in PR and León contain all seven types of FAP. Those measured in PR extend out to 10 μm,
regardless of type. In León, at EODs < 2 μm, the number concentrations are always higher in
concentration than those in PR but never exceed 7 μm in size. The arrival of the AD significantly
changes the shapes of the size distributions, especially those in León, by bringing FAP that extend
out to > 10 μm. The primary impact on the PR aerosols is to increase their number concentrations
across all sizes and FAP types, while making little changes in the maximum EOD, except for the FAP
Type B whose maximum EOD increases from 8 to 10 μm. The change in the size distributions of
Type AC (Fig. 8e) with the arrival of the dust is particularly noticeable in PR and León. During non-
AD periods the concentrations of Type AC FAPs is quite low at both measurement sites and then the
arrival of AD increases the concentrations by several orders of magnitude, suggesting that the dust
FAPs vary from the normal, background FAP in concentrations, size and types.





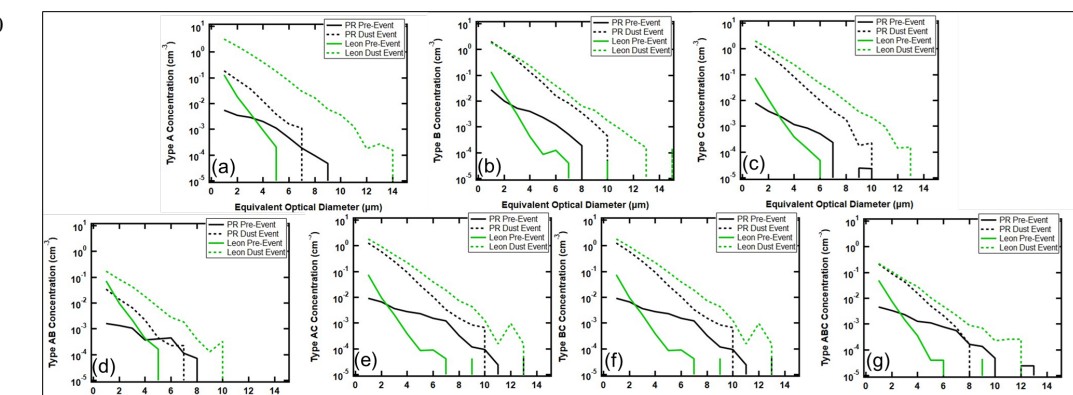

*Figure 8. Average number concentration size distributions of FAP in PR (black) and León (green), before (solid) and during (dashed) AD events for (a) Type A, (b) Type B, (c) Type C, (d) Type AB, (e) Type AC, (f) Type BC, (g) Type ABC.*

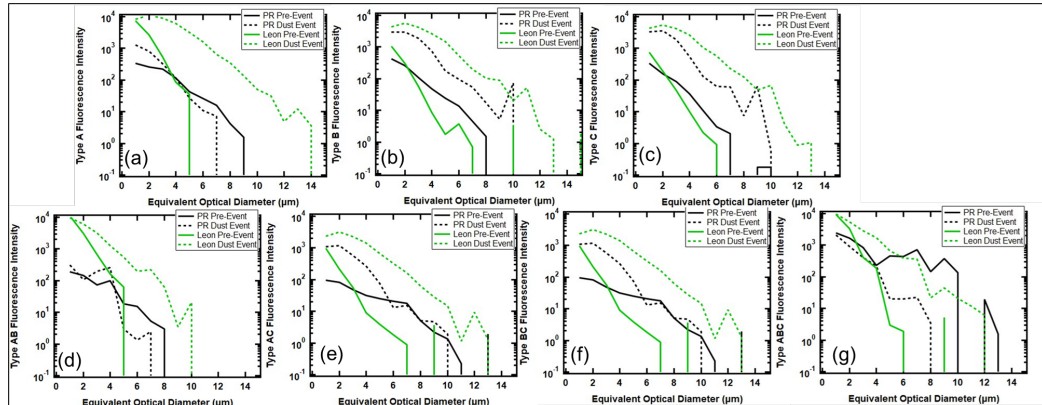

*Figure 9. Average fluorescence intensity of size distributions in PR (black) and León (green), before (solid) and during (dashed) AD events for (a) Type A, (b) Type B, (c) Type C, (d) Type AB, (e) Type AC, (f) Type BC, (g) Type ABC.*

Similar to Fig. 8, Fig. 9 illustrates the average fluorescence intensity as a function of size for the seven FAP types. Keeping in mind that the average fluorescence intensity is unrelated to the average number concentration, we observe that the average fluorescence intensity of the León pre-dust aerosols are greater than those in PR in the size range less than 2 µm, i.e., the difference in fluorescence intensity below 2 µm is not a result of higher concentrations in León, but possibly a different type of FAP. Similar to the comparison of the number concentrations, the PR FAP extends out beyond 8 µm for all types. The size distributions of Types B and C aerosols measured in PR and León, pre-dust, are quite similar in shape whereas the León size distributions are quite different from those in PR for the other types, suggesting a dissimilar population of bioaerosol taxa at the two locations. The arrival of the dust leads to shifts in the size distributions that are similar for the PR and León Types B and C; however, the León fluorescence intensity increases by more than two orders of magnitudes while the PR intensities are about a factor of 10 higher in magnitude. Whereas the León intensities of all FAP types broaden from a maximum of 6 µm out to more than 10 µm, the PR distributions show little broadening except for Types B and C. The primary difference between the pre-dust and dust events in PR is an increase in intensity of FAP < 5 µm as compared to the increase in intensity over all sizes with the León distributions. The difference between the PR and León changes in size distributions with the arrival of the AD is particularly striking for the Type ABC aerosol. The León distributions broaden from a maximum of 6 µm to 12 µm, the PR distributions narrow from 10 µm to 8 µm and





the average fluorescence intensity decreases over this size range by more than a factor of 10. These
contrasts between the PR intensity size distributions with those of León provide an additional piece
to the puzzle associated with how FAP are mixed with AD when the plumes reach the respective
locations.
A comparison of the shape factor size distributions, shown in Fig. S4, informs us that FAP types A
and AB are quasi-spherical (fluorescence intensity < 15%) while the other FAP types are more
aspherical (>15%) at EODs between 6 µm and 10 µm. There is not a significant difference between
PR and León FAP, either pre-dust or during the AD events.
Figure 10 highlights the transitions in the size distribution shapes, for all FAP types, by comparing
the MEOD metric derived for all dust and no-dust periods, similar to what was shown in Figs. 7d and
e. In this case, however, the MEODs were extracted from the size distributions of FAP number
concentration (Fig. 10a) and fluorescence intensity (Fig. 10b). There is a stark difference seen
between the background MEODs of number concentration and fluorescence intensity when
comparing the background (no-dust) values from PR and León. The MEODs range between 5 µm
and 8 µm in PR while the León MEODs are much smaller, between 2 µm and 5 µm. The second
major difference between the PR and León MEODs is that the PR MEODs decrease with the intrusion
of dust, with all FAP types except B and C, while the MEODs increase over all the FAP types. These
differences were reflected in the size distributions where we see significant increases in the number
concentration and fluorescence intensity of the FAP < 5 µm in PR whereas it is the concentrations
and intensities of FAP > 5 µm that increase in León.

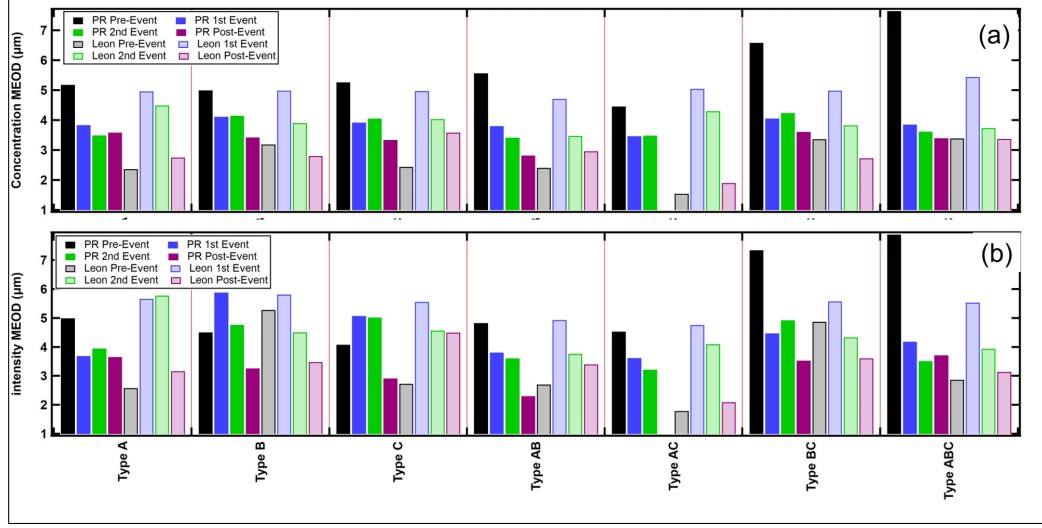

*Figure 10. (a) Average median equivalent optical diameters (MEOD) of the size distributions of the number*
*concentrations for the seven FAP types. The color coding delineates the locations (PR and León) and dust event*
*conditions (before, during and after). (b) Same as (a) except the MEODs are from the size distributions of the average*
*fluorescence intensity.*

**3.3   Complementary Meteorological and Aerosol Observations**

An evaluation of the meteorological state parameters and winds saw no indication of the dust arrival
in PR or León, i.e., we observe no significant difference in temperature, relative humidity, wind speed
or wind direction. Hence, the meteorological properties of the dust layer do not appear to have a
noticeable impact on the local meteorology in PR or León (Fig. S5 and S6).




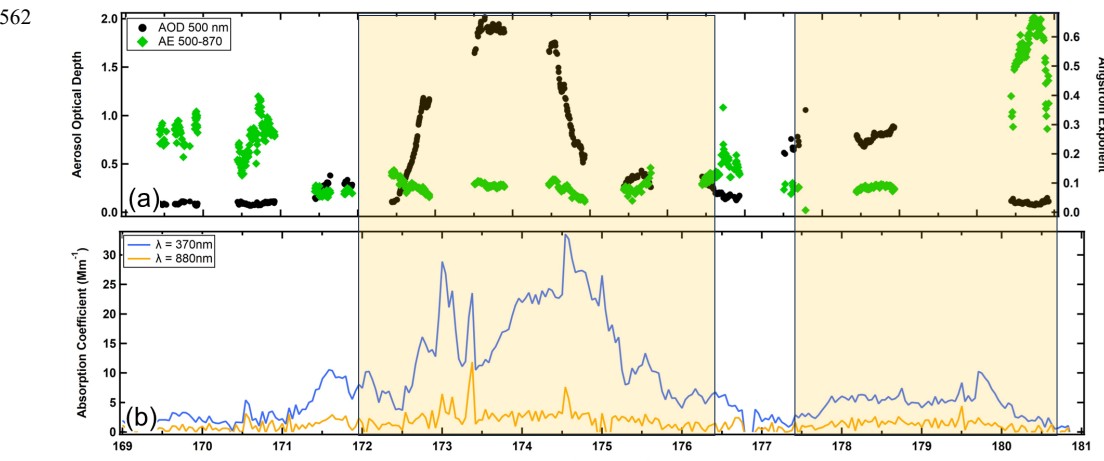

*Figure 11. Time series in Puerto Rico of (a) the aerosol optical depth (AOD) at 500 nm wavelength (black markers),*
*Ångström exponent derived from the 500 nm and 870 nm AODs (green markers) and (b) Absorption coefficients at*
*wavelengths of 370 nm (blue) and 880 nm (orange). The shaded areas demarcate the time periods when AD was in*
*the region.*
Figure 11 illustrates the impact of the AD on the aerosol optical properties in PR where the shaded
regions delineate the time periods with AD. In PR the trends in the extinction coefficients (Fig. 11b,
blue and orange curves) suggest that the leading edge of the AD layer might have already arrived at
the measurement site a day earlier than the measurements from the WIBS indicate (Figs. 4a and 5a).
The 370 nm extinction coefficient shows an increase on DOY171, reaching a peak in the middle of
the day before decreasing in the evening. The 880 nm extinction coefficient does not show the same
trend because dust absorbs at 370 nm and very little at 880 nm, although the light scattering at 880
nm produces a lower intensity extinction coefficient. There were no increases in wind speed or shifts
in wind direction (Fig. S1) that could indicate that these might be anthropogenic in origin, or possibly
local dust. This pattern is also reflected by a small increase in the aerosol optical depth (Fig. 11a,
AOD, black markers), which follows the same trend. The AOD, measured with a sun photometer,
can't distinguish the actual altitude where these new particles might be located; hence, these could be
dust particles that had been transported into the boundary layer where they would be measured by the
MET-1 OPC. The main body of the AD layer, identified from the WIBS measurements (Figs. 4a and
5a) arrived on DOY172, where it is also seen clearly in the 370 nm extinction measurements (Fig.
11b) and the AOD (Fig. 11a). Note that the AOD and 370 nm extinction coefficients, although
roughly correlated in time, will not follow the same trends if dust in the boundary layer and free
troposphere is arriving with a different periodicity than the dust that is sedimenting or being
transported downward by larger scale eddies. The other aerosol parameter plotted in Fig. 11a is the
Ångström exponent derived from the 500 nm and 870 nm AODs. This parameter is roughly inverse-
related to the average, median size of the aerosol particles. We observe in Fig. 11a (green markers),
that during periods with no-dust, the exponent is larger than during periods of dust, an expected result
given the significant increase in average EOD that was observed from the WIBS measurements.





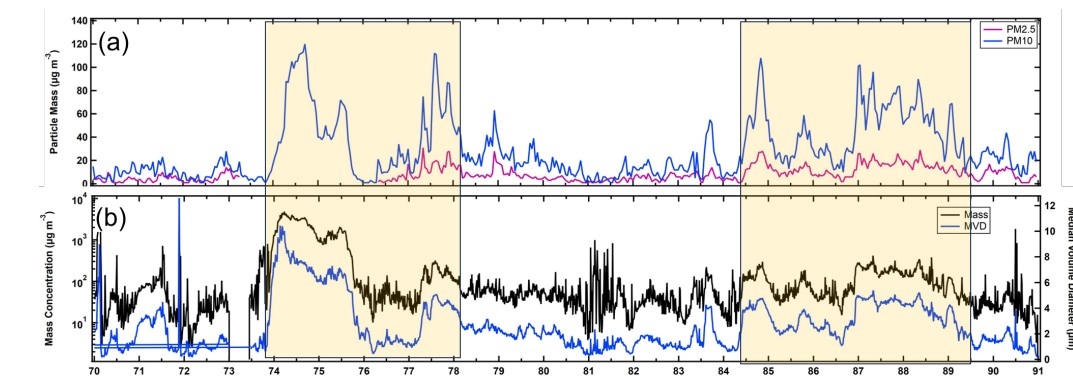

*Figure 12. Time series in León of (a) PM2.5 (magenta) and PM10 (blue) and (b) particle mass concentration (black curve) and median volume diameter (blue) measured with the FM120 in León. The shaded areas demarcate the time periods when AD was in the region.*

The AD incursion over León is reflected in the PM2.5 and PM10 measurements (magenta and blue curves, respectively) shown in Fig. 12a. Unlike the trends in the PM measured in PR, there is a clear periodicity in León where daily peaks are observed on most days, regardless if the AD is present; however, during the AD, the maximum PM is four to ten times larger than when the AD isn't present. Given that there doesn't appear to be any correlation with meteorological parameters, the trends are likely the result of changes in the depth of the boundary layer. As this layer grows during the day, due to radiative heating, the AD that is aloft in the free troposphere mixes downward and increases the PM near the surface. Figure 12b shows the mass concentrations derived from the FM120 size distributions (black curve) and the median volume diameter (blue curve). This complementary set of measurements, independent of the WIBS or air quality PM measurements, is highly correlated with the results from both instruments and show that the MEOD increases from < 3 µm to > 5 µm when the AD arrives. The very large mass concentrations are a result of the particles > 10 µm, as can be observed in the time series of the FM120 size distributions (Fig. S7). Between DOY 74 and 75 the size distribution is clearly bimodal with one peak at 5 µm and the other between 20 and 30 µm. These large particles are what drive the very high PM values seen in Fig. 12b, which are much larger than registered by the PM10 sensors in the air quality station.

## 3.3 Hirst Sampler Observations

The timeseries of micro and macroconidia fungal spore concentrations, collected in PR and sorted by size range, are drawn in Fig. 13. The microconidia < 3 µm (black bars) are always the highest in concentration, followed by the microconidia > 3 µm and microconidia < 10 µm (blue) and the macroconidia > 10 µm (green). The shaded regions are highlighting the periods of AD. There are differences in the concentrations of these fungal spores when comparing periods with and without AD, but they are subtle. Given that the dust plume mixes with the ambient aerosols *a priori* we have no reason to expect the spore concentrations to increase or decrease. More significant is the appearance of spore types that were not identified during the no-dust periods. Table II provides more explicit detail regarding the redistribution of spore types. More important than the total number concentrations are the appearance of new spore types and disappearance of others during the AD episodes. These are highlighted in the table, blue when periods of AD lack spores during no-dust periods, and red when spore types appear that were not in the no-dust periods. In addition, cells in the table are shaded orange when a spore type increases by ≥ 100% from no-dust to dust.






*Figure 13 Time series of spore concentrations, stratified by size, in Puerto Rico. The shaded regions are periods of AD*
*inundation.*

There are two spore types, Dreshlera Helmitosporum and Fusarium that were measured on the no-
dust days but were no longer identified during AD. Likewise, with the arrival of the dust, five new
spores appeared that were not previously seen in the background environment: Erysiphe/Oidium,
Periconia, Spegazzinia, Tetrapyrgos and Chaetomium. Of these, the Erysiphe/Oidium had the highest
concentration, four times higher than the others.

In León during March 2022 a total of 9 pollen types were identified. Cupressaceae and Populus, both
in their main pollen season (MPS), were the most abundant types (abundance relative: 43% and 40%,
respectively). The other pollen types presented relative abundance values lower than 5%. Some pollen
types such as Alnus, Corylus, Fraxinus and Ulmus were finishing their MPS, whereas Platanus,
Poaceae and Pinus were starting it. Salix was in the MPS during this period, although it is not an
abundant pollen in the ambient atmosphere. During the two AD intrusions, an increase in pollen
concentration compared to the previous day (DOY 75: >1000%; DOY 85: 300%) was registered
(Figure 14). During the first one, most of the counted pollen belonged to Cupressaceae. Nevertheless,
during the second AD inundation the predominant pollen was Populus. Days with AD inundation did
not show differences in airborne pollen diversity compared to days without AD intrusion.


*Figure 14. Time series of daily average pollen concentrations in León during the selected period. The yellow shaded*
*regions indicate the AD inundation.*



Table II

Number Concentration (m⁻³) of Fungal Spores in Puerto Rico (Maximum daily values)

| Macroconidia >10 µm | No AD | AD | Change % | Microconidia 3-10 µm | No AD | AD | Change % | Microconidia <3 µm | No AD | AD | Change % |
|---|---|---|---|---|---|---|---|---|---|---|---|
| Hifas fragmentos | 64 | 48 | -25 | Curvularia | 16 | 80 | 400 | Ascosporas | 3844 | 1621 | -58 |
| Cercospora | 207 | 96 | -54 | Dreshlera Helmitosporum | 16 | 0 | NaN | Basidiosporas | 7878 | 5051 | -36 |
| Helicomina | 48 | 80 | 67 | Erysiphe/Oidium | 0 | 48 | NaN | Cladosporium | 1462 | 271 | -81 |
| | | | | Fusarium | 16 | 0 | NaN | Chaetomium | 0 | 16 | 0 |
| | | | | Ganoderma | 302 | 207 | -32 | Coprinus/Agaricus | 128 | 128 | 0 |
| | | | | Leptosphaeria-Like | 32 | 48 | 50 | Diatrypaceae | 2034 | 1938 | -5 |
| | | | | Periconia | 16 | 16 | 0 | Smut/Myxomycete | 16 | 64 | 300 |
| | | | | Pithomyces | 16 | 64 | 300 | | | | |
| | | | | Pleospora | 64 | 16 | -75 | | | | |
| | | | | Nigrospora | 32 | 32 | 0 | | | | |
| | | | | Rusts Puccinia | 32 | 64 | 100 | | | | |
| | | | | Periconia | 16 | 16 | 0 | | | | |
| | | | | Spegazzinia | 0 | 16 | NaN | | | | |
| | | | | Ulocladium | 16 | 16 | 0 | | | | |
| | | | | Tetrapyrgos | 0 | 16 | NaN | | | | |

Regarding the analysis of bi-hourly pollen concentration (Figure 15) it can be observed that days with
AD presented the highest pollen concentrations from 2000 to 2400 UTC, which suggests pollen
transport from emission sources far away from the monitoring station. In addition, airborne fungal
spore taxa did not show significant concentrations during these days. The spore taxa identified during
the selected period were Cladosporium, Alternaria, Pleospora, Tilletia and Leptosphaeria.

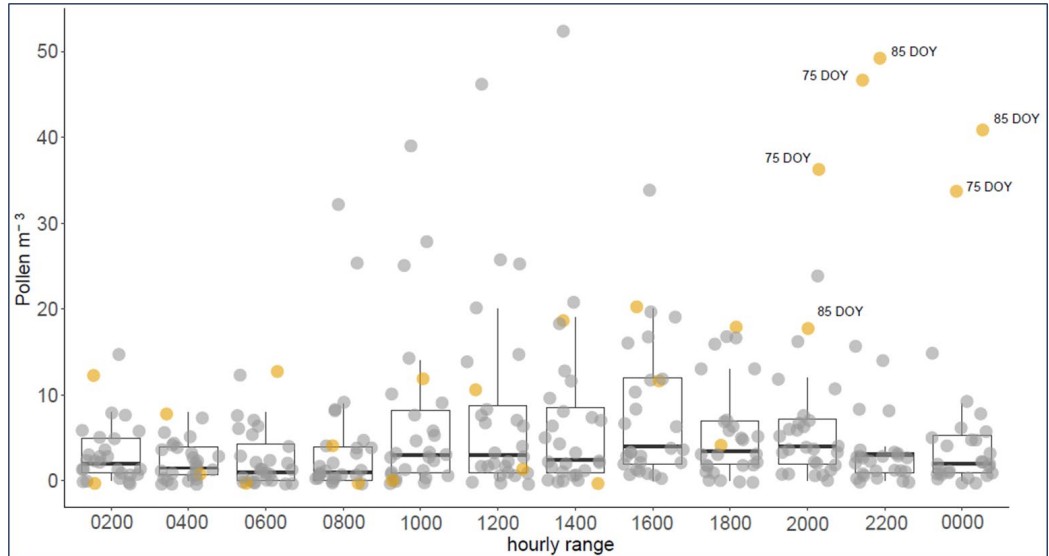

*Figure 15 Bi-hourly values of total pollen concentration recorded by Hirst trap for the selected period in León. Days*
*without AD inundation (●), Days with AD inundation (●).*



### 3.4 Back Trajectory Analysis

The origins and histories of the air masses were evaluated using the National Oceanic and Atmospheric Administration (NOAA) Hybrid Single-Particle Lagrangian Integrated Trajectory model (HYSPLIT) back trajectory model, incorporating the Global Data Assimilation System (GDAS) with one degree resolution (Stein et al., 2015; Rolph et al. 2017). The model was run for thirteen and five days for PR and León, respectively, time periods commensurate with the number of days between when dust was seen in the satellite data to originate over northern Africa and arrive at the two destinations, respectively. The ending altitudes were chosen to be 100, 500 and 1000 m based on previous studies that have shown that the AD layers can range in thickness between 100 and 1000 m (Ramírez-Romero et al., 2021). Figure 16 shows representative back trajectories for PR on June 22, 2020, color coded by altitude and with markers (red) that indicate when and where the air was within the mixed layer and when the air mass encountered precipitation (light blue markers). These mixed layer parameters were selected to show where the originating air might have first picked up the dust and then later where the air might have interacted with other sources of aerosols, e.g. marine aerosols when passing over the Atlantic Ocean. The precipitation is added because it can contribute to cloud processing of aerosols and potential removal of particles before the air arrives at its destination.

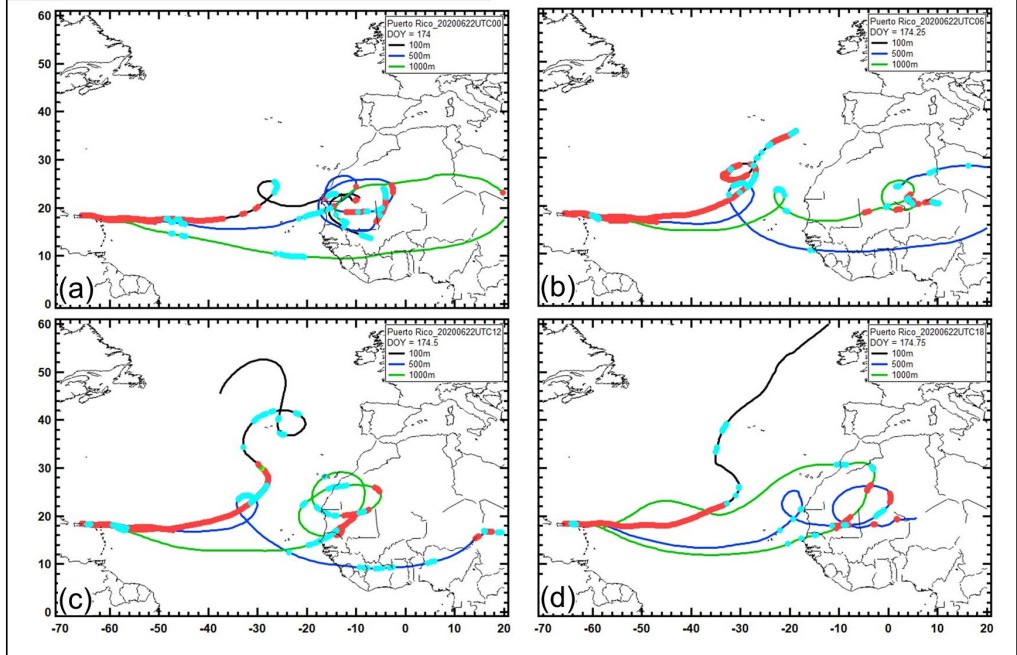

*Figure 16. Thirteen day back trajectories of air masses arriving at 100 m (black curve), 500 m (blue curve) and 1000 m (green curve) over Puerto Rico. The red markers show every hour the air was in the mixed layer and the light blue markers denote each hour where rain was encountered. These are from June 22, 2020 at (a) 0000 UTC, (b) 0600 UTC, (c) 1200 UTC and (d) 1800 UTC.*



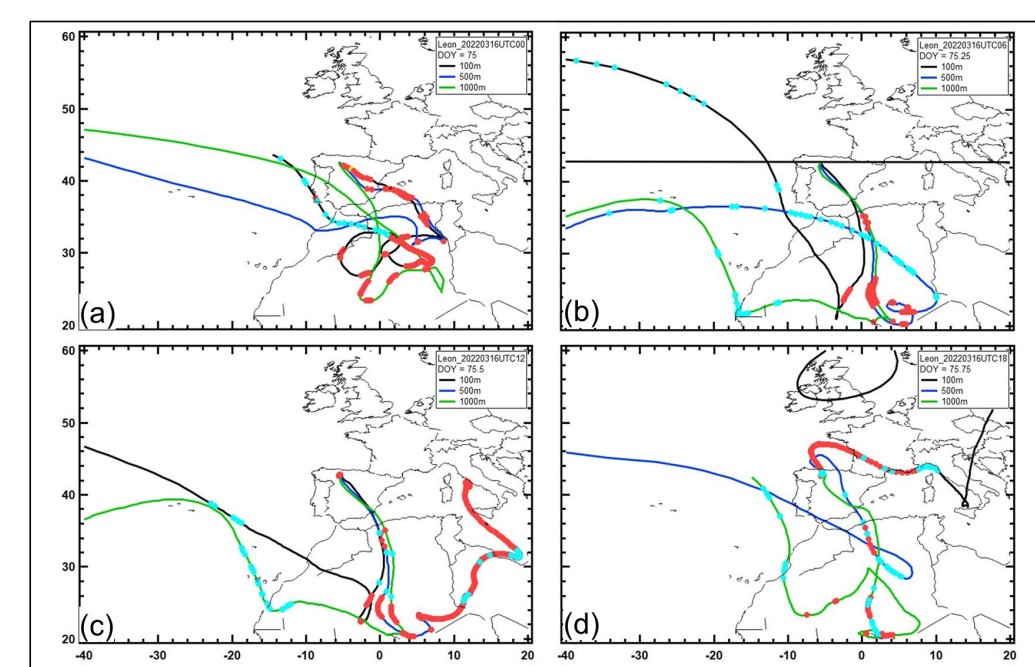

*Figure 17. The same as Fig. 16 but for five day back trajectories of air masses arriving over León on March 16, 2022 at (a) 0000 UTC, (b) 0600 UTC, (c) 1200 UTC and (d) 1800 UTC.*

Over the 24-hour period that AD was arriving on June 22nd, at UTC 0000, 0600, 1200 and 1800 (Figs. 16a-d) the air can be seen originating from over the African Sahara and Sahel. At UTC 0000 all three trajectories had been over this region and the red markers also show that they had been there in the mixed layer at different times, confirming that particles indigenous to that region would have originated there. At 0600, 1200 and 1800 UTC the 100, 500 and 1000 m trajectories do not always indicate being in the mixed layer, but at least one of them does; hence, the AD continues to be transported to PR over these time periods. It is also important to note that the 100 m trajectory, as well as sometimes the 500 m trajectory, arrive over PR after traveling several hundred kilometers (> 24 hrs) in the mixed layer. With respect to cloud processing the HYSPLIT model indicates that throughout the day the air had encountered precipitation first over Africa and then on its travel over the Atlantic Ocean before arriving in PR.

Figure 17 provides the same information for León over four five-hour time periods on March 16, 2022. Similar to what was observed with the air masses that brought AD to PR, the air masses that arrived over León at 100, 500 and 1000 m had all been in the mixed layer in northern Africa for varying lengths of time. Whereas most of the AD that arrived over PR originated in western Africa, those air masses over León were bringing particles from regions in northern and northeastern Africa. Much of the air, particularly that which arrived at 500 m over León, had also encountered frequent periods of precipitation as indicated by the model.

## 4.0 Discussion

In Section 2.3.3 we posed questions related to how the WIBS measurements could be used to distinguish differences in bioaerosol taxa in the background FAP of PR and León and between background and dust events. Although the size distributions of the number concentrations and intensities of the FAP Types in PR and León cannot be used to speciate bioaerosols, the distinct differences in the relative fraction of total FAP in smaller and larger particles indicate that the



mixtures of BSP types, i.e., bacteria, mold or pollen, are clearly dissimilar. This is observed when
comparing the two regional background aerosols and when comparing the changes when AD arrives.
The contrasts in FAP properties are highlighted by placing their physical and fluorescing properties
in the context of these same properties generated using laboratory studies, as was demonstrated in
Fig. 2. Figures 18 and 19 summarize the FAP properties for the PR and León regions, before, during
and after dust events as they compare to the FAP properties of bacteria, mold and pollen measured in
the laboratory. A cursory examination of these two figures confirms that the FAP properties are
significantly different between PR and León, without and with dust. While this result should not be
considered surprising, displaying the FAP properties as illustrated in these figures offers a way to
indirectly compare the ambient FAP properties with those of actual BSPs. The distribution of FAP in
the PR background aerosol (Figs. 18a and d) corresponds mostly to the laboratory mold and pollen
with only a small fraction falling into the bacteria type and size. The PR FAP that falls in the pollen
region is mostly in EODs < 5 µm. There is a population of FAP types B, C and AB that are found in
the ambient environment but have no corresponding laboratory BSP types that they can be associated
with. With the arrival of the dust, the FAP maps shift significantly with the largest majority of the
fluorescing particles appearing in the Type B, C and BC categories and at EODs < 5 µm. This is a
distinct shift in FAP types caused by the arrival of dust.

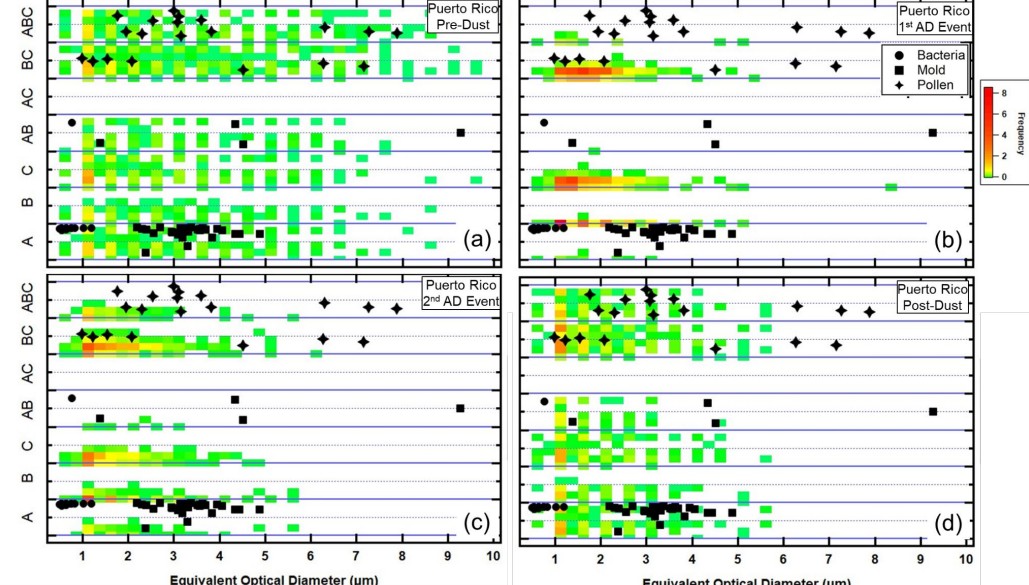

*Figure 18. Similar to Fig. 2. Except the laboratory BSP maps are combined with the frequency of occurrence (color*
*coded) of FAP types. The color scale denotes how frequently during the two day periods the FAP types and EODs fell*
*within the different regions. (a) Puerto Rico two days before the AD event, (b) and (c) Puerto Rico during the 1st and*
*2nd AD events and (d) Puerto Rico after the AD event.*






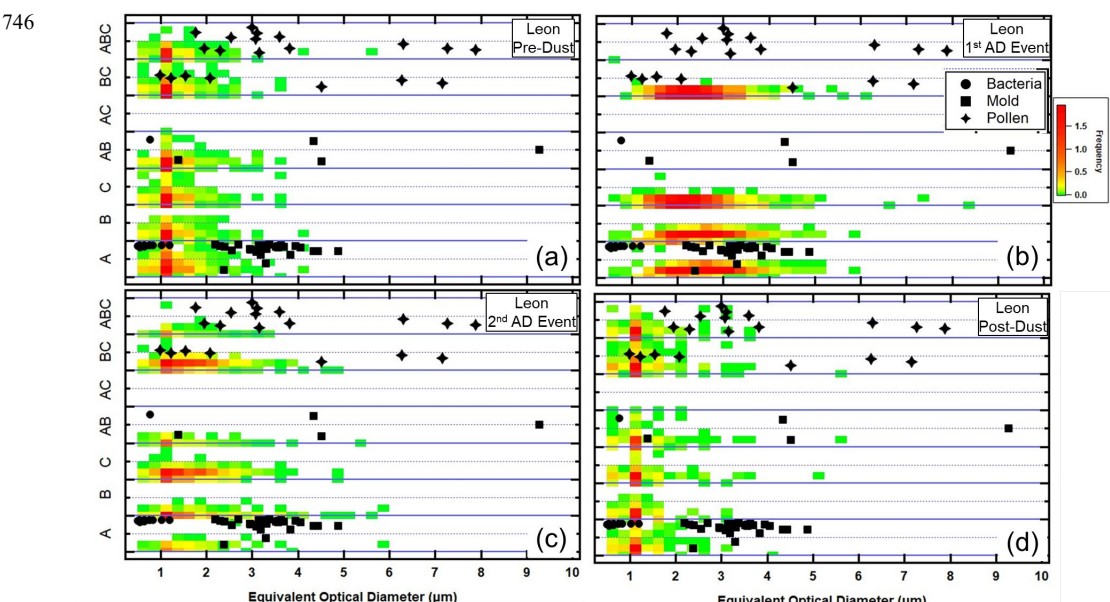

***Figure 19. The same as Fig. 18, except for measurements in León.***

The FAP patterns in the León aerosols are shown in Fig. 19 and suggest that the largest number of
background aerosols should be considered bacteria-like and small, pollen-like as compared to the
laboratory BSPs. The highest frequencies are found at EODs < 2 μm, evenly distributed over all FAP
types except Type AC. This suggests that many of the FAP measured in León are different from the
laboratory BSP taxa. The arrival of the dust dramatically shifts the pattern of the FAP – the highest
frequencies are now in Types A, B, C and AC and the EODs are now centered between 4 μm and 6
μm during the first AD event. These sizes decrease during the 2nd AD event before returning to mostly
< 2 μm in the post-dust time period. These results suggest that the background FAPs are mostly
bacteria and pollen, similar to the background air, with some fraction with FAP types that do not
correspond to the taxa of the BSPs tested in the one laboratory study. The dust brings in bioaerosols
whose FAP properties include those that are similar to bacteria, mold and pollen, but now with larger
EODs.

The results shown in Figs. 18 and 19 offer compelling evidence that the WIBS measurements
distinguish between types and compositions of bioaerosols and that the no-dust and dust cases can be
clearly separated, as can the background populations of PR and León. What about information on
how FAP are physically mixed with non-FAP aerosol? To address this question, we remind the reader
what the differences between internally and externally mixed aerosol ensembles are. Figure S8 is a
conceptual diagram that compares internally and externally aerosols. In short, we do not expect to
have only one or the other type of situation given that turbulent mixing will lead to the eventual
combination of the two. Nevertheless, there are several reasons to expect that one or the other might
dominate, depending upon the age of the air masses and the types of physical processes that can occur
between the origin of the dust plume and when it arrives several hours or days later in PR and León.

Referring back to Fig. 10a we observe a clear difference between PR and León when comparing the
changes in the MEODs, over all FAP types, when dust arrives. The MEODs ***decrease*** by 20-30% in
PR and ***increase*** by 30-50% in León. These opposite changes were also seen in the size distributions
shown in Figs. 8 and 9, i.e., in general, FAP concentrations in sizes <5 μm increased in PR and
decreased in León. These differences can be explained, to some degree, by differences in the mixing
state of FAP and dust during AD events. The large decrease in the FAP MEOD in PR suggests that



the dust is bringing many more small FAPs than are found in the background; however, the increase
in the larger FAPs indicates that some of these FAP are also internally mixed with the larger dust
particles. Hence, the PR dust may be more externally than internally mixed (Fig. S8a). On the other
hand, the AD event in León increased the MEODs over all the sizes; however, the larger particles (>5
μm) increased proportionately more than those <5 μm. This leads to the conclusion that the FAP in
the León AD event are more internally mixed, i.e., that a large fraction of the AD FAP are attached
to or mixed with the AD, as illustrated in Fig. S8b.
If we assume that the AD that arrives in PR and León originated through similar processes over
northern Africa, whereby dust and FAP are lofted from the surface, then the differences that are
observed in the FAP properties when they arrive in León and PR are likely a result of the
transformations that occurred during their transport. The three primary processes that lead to these
transformations are coagulation, sedimentation and precipitation. The AD had traveled over much
longer distances and time before reaching PR than when arriving over León. The back trajectory
analyses showed that these air masses had also traveled many hours in the boundary layer prior to
reaching PR. Some of the FAP are likely attached to dust particles when they are lifted from the
surface at their origin, while others will collide with the dust during transport as a result of small scale
turbulent eddies, and sedimentation of the larger dust particles falling through the smaller FAP.
Electrical charging of the particles, leading to further coagulation cannot be discounted. Particles with
an aerodynamic diameter of 1 μm and a density of 2 g cm$^{-3}$ fall at a speed of 6 m/day while a 10 μm
particle falls at 500 m/day; hence, the particles $\geq$ 10 μm will fall 2.5 km and 6.5 km during their 5
and 13 day travel from Africa to León and PR, respectively. This type of removal of the larger
particles, while smaller particles remain aloft, can explain the difference between PR and León
mixtures of FAP and dust. Not only did the particles in the AD air masses that arrived in PR have
three times longer to fall than those in León, but the back trajectory analysis also revealed that the air
arriving in PR had been in the mixed layer many more hours than the air masses reaching León.
Traveling in this layer would place the particles much closer to the surface and have a shorter distance
to sediment and be removed.

## 5.0 Data availability

The WIBS data and complementary aerosol measurements described in this manuscript can be
accessed at the Zenodo repository, under DOI .10.5281/zenodo.10680977 (Baumgardner, 2024)
.

## 6.0 Summary and Conclusions

Two major African dust events, one over the island of Puerto Rico and the other over the city of León,
Spain have been analyzed, the former in June, 2020 and the latter in March, 2022. From measurements
with two Wideband Integrated Bioaerosol Spectrometers (WIBS) and complementary aerosol data
we make the following observations and conclusions:

1. The intrusion of dust over the Caribbean and Iberian Peninsula leads to a significant impact on the size distributions and composition of the local populations of aerosols.
2. Differences in the FAP sizes and fluorescing properties, prior to the AD events, are clearly seen in comparisons between the background aerosol populations in PR and León.
3. The arrival of AD over the two regions significantly alters the properties of the local aerosol populations as observed in the WIBS and PM measurements. The magnitude of these altered properties are different at the two locations, differences attributed to the age of the AD air masses, five and 13 days old, when arriving in León and PR, respectively.
4. As deduced from changes in the shapes of the FAP size distributions, with the intrusion of the AD the FAP is both internally and externally mixed with other non-FAP particles in the dust



plume; however, the AD that arrives in PR appears to have a much higher proportion of
externally mixed FAP than León.

5. The comparison of the maps of relative frequency of FAP Types and their average EOD,
juxtaposed with laboratory bacteria, mold and pollen, indicates that the mixtures of FAP and
dust in PR are significantly different than those in León. The AD dust over PR clustered most
in FAP types C and BC while in León the primary AD types were A, B, C and AC. When
compared with the laboratory FAP, Type A is related to bacteria and BC to pollen. Types B
and C were not common in the laboratory measurements used in this study (Hernandez et al.,
2016) nor did other similar laboratory studies, e.g., Savage et al. (2017), have these types of
FAP.

The analysis approach that has been introduced in this study highlights the importance of using
metrics that focus on relative changes in the number concentration and fluorescence intensity size
distributions of the seven types of FAP. The median equivalent optical diameter (MEOD) is a
sensitive metric that can quantitatively document these changes along with maps of the frequency of
FAP type versus EOD that highlight how the FAP types in AD are significantly different from
background FAP in PR or León.
These two data sets will be a useful contribution to the larger data bases of African and Asian dust
aerosols that have been transported large distances and that may be carrying bioaerosols, some which
may be similar to those found in the local regions inundated by this dust while other might be more
damaging to the environments where they eventually are deposited or inhaled.
**7.0   Competing Interests**
The contact author has declared that none of the authors has any competing interests.
**8.0   Acknowledgements**
The authors gratefully acknowledge the NOAA Air Resources Laboratory (ARL) for the provision
of the HYSPLIT transport and dispersion model and/or READY website
(https://www.ready.noaa.gov) used in this publication. This work was partially supported by the Junta
de Castilla y Leon co-financed with European FEDER funds (Grant LE025P20), by the
AEROHEALTH project (Ministry of Science and Innovation, co-financed with European FEDER
funds. Grant PID2019-106164RBI00) and by National Science Foundation-MRI grant (1829297).
Furthermore, it is part of the project TED2021-132292B-I00, funded by
MCIN/AEI/10.13039/501100011033 and by theEuropean Union "NextGenerationEU"/PRTR.
**9.0   Author contributions**
B. Sarangi and B. Bolaños-Rosero provided all the data from the Puerto Rico site, A. Calvo and R.
Fraile provided the WIBS and FM-120 measurements from the León, Spain measurement site. D.
Baumgardner and D. Hughes assisted in the processing of WIBS measurements from PR and León,
A. Rodríguez-Fernández and D. Fernández-González provided the Hirst sampler data from León, C.
Blanco-Alegre, C. Gonçalves and E. D. Vicente operated the WIBS and FM-120 during the León
project, O. L. Mayol Bracero helped to edit the manuscript and M. Hernandez contributed the
laboratory studies of FAP.

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
