# Peer review of "Measurement Report: Comparative Analysis of Fluorescing African Dust Particles in Spain and Puerto Rico"

_EGUsphere, 2024_

## Author Response (AR1)

Response to Reviewer #1

Review of "Measurement Report: Comparative Analysis of Fluorescing African Dust Particles in Spain and Puerto Rico" by Bighnaraj Sarangi et al.

Recommendation: Minor Revisions

The manuscript "Measurement Report: Comparative Analysis of Fluorescing African Dust Particles in Spain and Puerto Rico" reports measurements taken at two sites during episodes of African dust using two Wideband Integrated Bioaerosol Spectrometers (WIBS), which show that bioaerosol-like fluorescing aerosol particles (FAP) that can be associated with these dust episodes. Satellite and back trajectory analyses confirm that dust from Northern Africa was the source of the particles during both events. The WIBS measures the size of individual particles in the range from 0.5 µm to 30 µm, derives a shape factor and classifies seven types of fluorescence from the FAP. Some new indicators are introduced to provide a more quantitative way to compare the FAP characteristics derived from particles measured under different environmental conditions. The analysis highlights the similarities and differences at the two locations and reveals differences that can be attributed to the age and history of the dust plumes.

In general, the paper is well written and presented in a logical way. It is a timely and important piece of work. I therefore recommend publication of this paper in Esphere after minor revisions. My comments are listed as follows:

Major Comments:

1. In Table 1, a lot of data is used in this paper. How does the author consider the differences between laboratory and field data in different spatial and temporal resolutions? How does comparison ensure accuracy?

Response: We appreciate this question from the reviewer; however, perhaps we don't understand the nature of the questions. Laboratory measurements of bioaerosols are designed to document the response of the WIBS to different taxa of bacteria, fungi and pollen. They are not meant to replicate what happens in ambient conditions. Although elevated humidity can cause plants to emit spores, as far as we know, there is no reason to think that the fluorescing characteristics of bioaerosols in the laboratory will differ from those in the ambient. The relative magnitudes of Type A, B or C fluorescence intensity for Colorado yellow pine pollen, for example, should be the same regardless of whether these are measured in the laboratory or ambient air. We understand that laboratory aerosol measurements are under controlled conditions as compared to field measurements which are rather more complex with respect to their temporal and spatial distribution of aerosols. As far as fluorescent aerosol measurements using WIBS are concerned, the lab measurements provide different taxa of bioaerosols or fluorescent aerosols representative of those in the ambient. Numerous airborne fluorescent aerosols have been tested in the laboratory (Hernandez et al., 2016; Savage et al., 2017) to provide databases of bioaerosol fluorescing aerosol particles that can be compared to ambient particles. That being said, however, we understand the point that the reviewer is making and have added extra verbiage to declare that we are making the assumption that a taxa of bioaerosol measured

in the laboratory will exhibit the same fluorescence characteristics as the same taxa in the field.

We have added this sentence in section 2.3.2 "Here we are making the assumption that ambient conditions like temperature and humidity do not change the fluorescence properties of a particular taxa from those measured in the laboratory for the same taxa."

2. Inline 443-444 of the manuscript, "The pre-dust size distributions of non-FAP aerosol (Fig. 6a) are almost identical at both sites, with a small fraction of the León particle population larger than those in PR." How does Figure 6a arrive at this conclusion without showing non-FAP concentrations?

Response: The reviewer makes a valid point. Figure 6a is the FAP+non-FAP whereas 6b is just the FAP. The total aerosol population is dominated by the non-FAP particle population, particularly during the dust event as size larger than 5 µm, as can be seen when comparing with the FAP-only size distribution shown in Fig. 6b. The large difference between the total and FAP distributions brings us to the conclusion that we have now expanded a bit to read:

" The pre-dust size distributions of all aerosol particles, non-FAP and FAP (Fig. 6a), are almost identical at both sites, with a small fraction of the León particle population larger than those in PR. The arrival of dust leads to almost two orders of magnitude increase in both the PR and León concentrations, over all sizes, and brings significant numbers of particles larger than 10 µm."

3. In Figure 11b, The ordinate in the figure is the absorption coefficient, and the following text explanation is the extinction coefficient, please unify the statement.

Response: Thanks to Reviewer for catching this error, which has now been corrected in the revised manuscript, replacing all references to "extinction" with "absorption".

4. Inline 364 of the manuscript, Leon is misspelled.

Response: Thanks to Reviewer for catching the typo error. This has been corrected.

5. PM in the manuscript pay attention to modify the subscript.

Response: We have now rectified the subscript error.

6. In Figure 15, abscissa 2400 is displayed incorrectly.

Response: Thanks to Reviewer for catching the error in the abscissa. We have corrected it in the revised manuscript.

7. In line 674 and 826 of the manuscript, the HYSPLIT model was run for thirteen and five days for PR and León, respectively. Why did the author choose these two time periods?

Response: As we described already in the manuscript, the satellite data was used to determine the amounts of time that it was taking for the dust to travel from Africa to Puerto Rico and León, respectively.

8. In line 828of the manuscript, as for figures 8 and 9, figures 18 and 19 show basically opposite results, which the author believes are the result of the mixing of aerosols internally and externally. This statement is very novel, please add some testimony.

Response: We would like to direct the reviewer to the section in the discussion where we address the question of internal vs external mixing. We copy it here as a reminder and think that we have already explained our reasoning clearly:

"Referring back to Fig. 10a we observe a clear difference between PR and León when comparing the changes in the MEODs, over all FAP types, when dust arrives. The MEODs decrease by 20-30% in PR and increase by 30-50% in León. These opposite changes were also seen in the size distributions shown in Figs. 8 and 9, i.e., in general, FAP concentrations in sizes <5 µm increased in PR and decreased in León. These differences can be explained, to some degree, by differences in the mixing state of FAP and dust during AD events. The large decrease in the FAP MEOD in PR suggests that the dust is bringing many more small FAPs than are found in the background; however, the increase in the larger FAPs indicates that some of these FAP are also internally mixed with the larger dust particles. Hence, the PR dust may be more externally than internally mixed (Fig. S8a). On the other hand, the AD event in León increased the MEODs over all the sizes; however, the larger particles (>5 µm) increased proportionately more than those <5 µm. This leads to the conclusion that the FAP in the León AD event are more internally mixed, i.e., that a large fraction of the AD FAP are attached to or mixed with the AD, as illustrated in Fig. S8b."

9. It is suggested to add references in introduction.

Recommend citation:

Wang, Y. et al.,2023: Identification of fluorescent aerosol observed by a spectroscopic lidar over northwest China, Optics Express, 31, 13.

Sugimoto, N. et al.,2012: Fluorescence from atmospheric aerosols observed with a multi-channel lidar spectrometer. Optics Express, 20 (70), 20800-7.

Citation: https://doi.org/10.5194/egusphere-2024-446-RC1

Response: We thank the reviewer for reminding us that lidar can also measure fluorescence and we have added the following to the introduction: "Fluorescence from dust particles has

also been detected with lidar, such as the measurements reported Sugimoto et al. (2012) and by Wang et al. (2023) who report relatively strong, broad fluorescence from Asian dust and air-pollution aerosols transported from urban and industrial areas at wavelengths between 343nm and 526nm."

Response to Reviewer #2

General comments:

The paper, although a little too long, presents interesting analysis of two dust event periods and the presence of bio-aerosols, one over Puerto Rico, and one over Leon (Spain). The first part of the paper, which describes measurements method, is probably the best part of the paper.

The second part, the analysis of the two dust event periods, is the main limitation of the paper: Why these two events only, which do not occur at the same year and period? We cannot speak of a comparison just from two episodes that have no reason to be similar or different. At least, it is just two cases studies, which are interesting by themselves. The authors must explain why they have considered only these events, how they can be compared or not, and how they can be representative of not of dust events in two different regions of the world.

Response: We respectfully disagree with the reviewer that two events or dust episodes cannot be compared and they are just two case studies. What the two events that we are comparing have in common is that the source of the dust is from the same regions of Northern Africa. We acknowledge that they are from different seasons, but this should not impact significantly the characteristics of the dust itself that is being lifted from the surface by strong winds. Those two events were chosen because of their historical significance. The June 2020 dust event over Puerto Rico was the highest dust event observed in the last 26 years based on the ground data available, in terms of the amount of dust transported and its geographical extent (Yu et al., 2021). Whereas Leon recorded the highest dust event in the 22 years (García Valero, Juan Andrés. "Report on the intrusion of dust of Saharan origin over the Spanish peninsular territory between March 14 and 16, 2022." (2022)) based on the air quality data available at that location. Although these two events did not occur at the same time, the magnitude of dust they carried from Northwest Africa to two different locations of different geography enabled numerous opportunities to study the properties of these aerosols. For example, a better understanding of dust transport from Africa to the Caribbean and the Iberian Peninsula, the influence of meteorology over dust transport, dust-bioaerosols link, dust-cloud interactions, validate the outcome of the dust forecasting models, etc.

Reference:

García Valero, Juan Andrés. "Report on the intrusion of dust of Saharan origin over the Spanish peninsular territory between March 14 and 16, 2022." (2022).

Yu, H., Tan, Q., Zhou, L., Zhou, Y., Bian, H., Chin, M., Ryder, C. L., Levy, R. C., Pradhan, Y., Shi, Y., Song, Q., Zhang, Z., Colarco, P. R., Kim, D., Remer, L. A., Yuan, T., Mayol-Bracero, O., and Holben, B. N.: Observation and modeling of the historic "Godzilla" African dust intrusion into the Caribbean Basin and the southern US in June 2020, Atmos. Chem. Phys., 21, 12359–12383, https://doi.org/10.5194/acp-21-12359-2021, 2021.

The title must be changed to specify that only two events are considered.

Response: We have changed the title in the revised manuscript.

   "A Comparative Analysis of an Intensive Incursion of Fluorescing African Dust Particles over Puerto Rico and Another Over Spain.

Also, one important information is missing: the altitude of the dust plume. Do all the particles of the plumes have fallen on the ground, or do they mainly travelled above the ground, or do only the lower part of the plume have touched the ground? The ground-based meteorological parameters could be not representative of the meteorological conditions a few km above the ground.

   Response: We have added a dust plume height for PR and Leon in the revised manuscript.

The figures below show the height of the air mass back trajectories drawn at three different altitudes (e.g., 100 m, 500m, and 1000 m) on the peak days of the dust event observed in PR and Leon. The air mass height reduced when it moved from Northwest Africa to Puerto Rico and Leon. It shows significant dust deposition at the locations under study (Record amount of $PM_{10}$ observed at both the sites).  As for the Caribbean (Puerto Rico), the historic African dust plume in the Caribbean was modulated by meteorology. The MEERA-2 meteorology associated with the dust episode, which focuses on geopotential height and wind vectors in detail, is discussed in Yu et al., 2021.

**NOAA HYSPLIT MODEL**
**Backward trajectories ending at 0000 UTC 25 Mar 22**
**GDAS Meteorological Data**

[Figure]

Source ★ at 42.59 N 5.58 W

Meters MSL

2171
1271



18 06 12 18 06 12 18 06 12 18 06 12 18 06 12 18 06 12 18 06 12 18 06 12 18 06 12 18 06 12 18 06
03/24 03/23 03/22 03/21 03/20 03/19 03/18 03/17 03/16 03/15 03/14 03/13 03/12

Job ID: 154644          Job Start: Sat Jul 22 00:39:01 UTC 2023
Source 1 lat.: 42.593100  lon.: -5.576000  hgts: 100, 500, 1000 m AGL

Trajectory Direction: Backward      Duration: 312 hrs
Vertical Motion Calculation Method:      Model Vertical Velocity
Meteorology: 0000Z 22 Mar 2022 - GDAS1

[Figure]

Figure: HYSPLIT air mass back trajectories height drawn for thirteen days over PR (a) and five days over Leon (b) on DOY 175 and 75, respectively.

Tables and legends of the Figure (and in particular for Figure 9) are often too small and thus difficult to read.

Response: We acknowledge the reviewer's concern and have added an additional, larger legend on Figures 8 & 9 for clarity. We also draw the reviewer's attention to the figure captions that also describe the nature of the lines, i.e. color and dashed/solid.

Thus, the paper needs a serious revision concerning its objectives and conclusions. Nevertheless the analysis is interesting and deserves publication after the revision.

Specific comments:

Figure 1 : It seems that that dots that are present in Figure 1b for AB type are missing in Figure 1a.

Response: We appreciate the reviewer comments concerning the error on Figure1a. This Figure has been corrected in the revised manuscript. The label for the AB types was inadvertently covering the squares that are mentioned by the reviewer.

Line 359: Most of the optical counter assume a given value for the refractive index. Obviously, the fog monitor considers the refractive index of water (with no imaginary part) to provide the size distribution of the particles. Obviously, the refractive index of dust particle is totally different and have an imaginary part. Thus the retrieved size distribution will be erroneous due to this calibration procedure. In fact, the dust particles at a given size will be less luminous than water droplets. Thus, the real size of the dust particles will be underestimated.

Response: We appreciate the reviewer's comment. We agree that FM-120 may underestimate the real size of the dust particles. In this study, the primary instrument used was WIBS, which was tested and calibrated using a standard polystyrene Latex sphere and is more accurate than FM-120 as far as the measurement of particle size distribution of dust is concerned. We have edited the manuscript now to read:

"The FM-120 was originally developed to measure fog droplet properties; however, the measurements are not specific to fog and in the presence of dust particles it will measure their size distributions but with a larger uncertainty because these particles will not be spherical. nor will they have a refractive index of water (1.33). The estimated uncertainty due to shape and refractive index uncertainty is approximately ±30%."

Line 625: Nothing is highlighted in blue in red in Table II, and the cells are no shaded.

Response: We thank the reviewer for catching this formatting error. However, color shading

is not allowed by the EGU journals. We are now using gray and black shades in table II the revised manuscript

Figure 14: The shaded regions are more in orange than in yellow.

Response: We thank the reviewer for noticing this. We have changed "yellow" to "orange in the caption.

Figure 15: What mean "DOY" preceded by a number?

Response: We apologize for the confusion of preceding "DOY" with the number instead of placing the number after DOY, i.e. DOY75 rather than 75 DOY. This has now been changed.

Figure 17: To what correspond the black vertical line in panel (b)?

Response: We thank the reviewer for identifying this error in Figure 17 (panel b). We have corrected it in the revised manuscript.

Figure S8 is unnecessary.

Response: Given reviewer comment, We have removed FigS8 in the revised manuscript.

Citation: https://doi.org/10.5194/egusphere-2024-446-RC2